Ceciamaralia, a new genus of Dorvilleidae (Annelida) from deep waters of the Southwest Atlantic Ocean and an insight into its relationship within the family

Bonaldo Rafael de Oliveira 1 2 3 rafael.o.bonaldo@gmail.com
Steiner Tatiana Menchini 2 4
http://orcid.org/0000-0002-6303-7244 Garraffoni André Rinaldo Senna 1 4
1 Laboratório de Evolução de Organismos Meiofaunais, Universidade Estadual de Campinas , Campinas, São Paulo , Brazil
2 Laboratório de Biodiversidade Bentônica Marinha, Universidade Estadual de Campinas , Campinas, São Paulo , Brazil
3 Programa de Pós-Graduação em Biologia Animal, Instituto de Biologia, Universidade Estadual de Campinas , Campinas, São Paulo , Brazil
4 Department of Animal Biology, Universidade Estadual de Campinas , Campinas, São Paulo , Brazil
Glasby Christopher
Electronic publication date: 2024 Oct 24
Publication date: 2024
Volume: 12
Electronic Location ID: e18358
Received 2024 May 3; Accepted 2024 Sep 28
Copyright: © 2024 Bonaldo et al.
Copyright year: 2024
Copyright holder: Bonaldo et al.
License: This is an open access article distributed under the terms of the Creative Commons Attribution License, which permits unrestricted use, distribution, reproduction and adaptation in any medium and for any purpose provided that it is properly attributed. For attribution, the original author(s), title, publication source (PeerJ) and either DOI or URL of the article must be cited.
License URL: https://creativecommons.org/licenses/by/4.0/

Keywords: Eunicida, ‘Polychaeta’, Morphology, New species, New genus, Taxonomy, Cladistics, Marine worms

Funding: Coordenação de Aperfeiçoamento de Pessoal de Nível Superior–Brasil (CAPES): 001 Conselho Nacional de Desenvolvimento Científico e Tecnológico-CNPq 301551/2019-7 São Paulo Research Foundation-FAPESP 2018/10313-0 This work was supported by the Coordenação de Aperfeiçoamento de Pessoal de Nível Superior–Brasil (CAPES–Finance Code 001), the Conselho Nacional de Desenvolvimento Científico e Tecnológico-CNPq (301551/2019-7) and the São Paulo Research Foundation-FAPESP (2018/10313-0). The funders had no role in study design, data collection and analysis, decision to publish, or preparation of the manuscript.

==============================
Dorvilleidae Chamberlin, 1919 is a family of Annelida containing some of the smallest ‘polychaetes’ species, being poorly studied worldwide, and with little knowledge regarding its diversity and occurrence. Samples obtained in oceanographic campaigns performed in the Southwest Atlantic Ocean (Brazilian coast) revealed a high number of specimens of dorvilleids, adding to our knowledge of the family’s biodiversity. A detailed morphological analysis of these organisms has revealed a new genus, Ceciamaralia gen. nov., with two new species. The new genus differs from other Dorvilleidae genera in (i) the robust and enlarged pharynx which are frequently everted, (ii) unique composition of maxillae, with an elongated pair of serrated basal plates and one pair of anterior free maxillary plates with a long and thin anterior spine and (iii) ventral cirri present only in few first chaetigers. Ceciamaralia lanai gen. et sp. nov. is characterized by the presence of a broad and large dorsal cirrus on a few anterior parapodia and by furcate chaeta in supra-acicular fascicles. While Ceciamaralia nonatoi gen. et sp. nov. presents one geniculate chaeta instead of one furcate, the absence of dorsal cirri and, in some specimens, the absence of palps. A cladistic analysis supported the monophyly of Ceciamaralia gen. nov. by four synapomorphies related to the unique morphology of its maxillae, pharynx and appendages. This study is part of several recent taxonomic studies aiming to elucidate and increase the knowledge of Dorvilleidae, since it is part of a Ph.D project focused on the family.

Introduction

The Order Eunicida (Annelida) comprises polychaetes that have an internal jaw apparatus composed of ventral mandibles and dorsal maxillae (Zanol et al., 2021). Dorvilleidae Chamberlin, 1919 encompass some of the smallest-bodied eunicid species. The family exhibits varied life-styles, from free-living worms to commensal and/or parasitic species, inhabiting unconsolidated and consolidated substrates, from the intertidal zones to great depths (Martin & Britayev, 1998; Martin & Britayev, 2018; Zanol et al., 2021).

Dorvilleidae is the only extant group of Eunicida that has a ctenognath-type jaw apparatus: two or four rows of symmetrical or subsymmetrical denticulate maxillary plates, upper comb-like jaws, and an unpaired posterior carrier-like structure (Zanol et al., 2021). Despite the small size of some dorvilleids, a great morphological heterogeneity among species is observed. Body appendages on the prostomium, parapodia and pygidium are important for the initial identification of species, presenting a diversity of sizes and shapes. The number and shape of chaetae and the internal jaw apparatus also show great morphological diversity, which is important for delimiting genera and species within the family (Paxton, 2009).

Currently, Dorvilleidae comprises about 200 species distributed in 32 genera, of which 13 are monotypic (Read & Fauchald, 2024), including the most recent described Ikosipodoides Westheide, 2000, while, almost ⅓ of the family species belong to the genus Ophryotrocha Claparède & Mecznikow, 1869. More than half of the Ophryotrocha diversity was described in the last 25 years (Read & Fauchald, 2024), as well as other studies encompassing biology, natural history, genetics and systematics (Zhang et al., 2023).

Relationships among genera and species of Dorvilleidae are also understudied. In a broad cladistic study (comprising all genera of the family known at the time), Eibye-Jacobsen & Kristensen (1994) analyzed relationships within Dorvilleidae using genera as terminal taxa, recovering several generic groupings. Other studies have focused on accessing the monophyly of some genera and their relationship with closely related genera; for example, the work by de Oliveira Bonaldo, Menchini Steiner & Zacagnini Amaral (2022) on Eliberidens Wolf, 1986a, which recovered the monophyly of the genus and discussed its morphological similarities with other genera. Other studies including morphological and molecular data of genera, like Parougia Wolf, 1986b (Yen & Rouse, 2020), Ophryotrocha (Kvalø Heggøy, Schander & Åkesson, 2007) and one focused on the monotypic parasitic species Veneriserva pygoclava Rossi, 1984, have analyzed relationships within other Dorvilleidae genera, mainly Ophryotrocha (Tilic & Rouse, 2024). These latter studies reveal that the scarcity of molecular data and viable specimens from which to extract such data are an obstacle to advancements in this field.

Among the reasons for the scarcity of knowledge on species of this family are: (i) the difficulty to perform sampling in deep waters, (ii) the rarity of some groups in the samples and (iii) the lack of taxonomists specialized in this group. The knowledge gap in Dorvilleidae systematics is worldwide and exemplified on the Brazilian coast, where, currently, there are only nineteen species recorded: Dorvillea angolana (Augener, 1918), Dorvillea moniloceras (Moore, 1909), D. sociabilis (Webster, 1879), Eliberidens forceps Wolf, 1986a, E. hartmannschroederae Hilbig, 1995, Meiodorvillea hartmanae de Oliveira Bonaldo, Menchini Steiner & Zacagnini Amaral, 2022, M. jumarsi de Oliveira Bonaldo, Menchini Steiner & Zacagnini Amaral, 2022, M. minuta (Hartman, 1965), M. penhae de Oliveira Bonaldo, Menchini Steiner & Zacagnini Amaral, 2022, Ophryotrocha puerilis Claparède & Mecznikow, 1869, O. zitae Miranda, Raposo & Brasil, 2020, Pettiboneia sanmartini Aguirrezabalaga & Ceberio, 2003, Pettiboneia sanmatiensis Orensanz, 1973, Protodorvillea biarticulata Day, 1963, P. kefersteini (McIntosh, 1869), Schistomeringos annulata (Moore, 1906), S. anoculatus (Hartman, 1965), S. longicornis (Ehlers, 1901), and S. rudolphi (delle Chiaje, 1828) (Amaral et al., 2006–2022); seven of them were recorded in three recent taxonomic studies (de Oliveira Bonaldo, Menchini Steiner & Zacagnini Amaral, 2022; de Oliveira Bonaldo et al., 2022; Miranda, Rodrigues & Brasil, 2020).

Recent oceanographic campaigns performed in the Southwest Atlantic Ocean (Brazilian coast) resulted in the collection of Dorvilleidae specimens, allowing for an increase in the knowledge on the biodiversity of the family in this region. By applying different methodologies, including light and scanning electron microscopy and cladistic analysis, we identified and described a new genus of Dorvilleidae, Ceciamaralia gen. nov. with two new species, Ceciamaralia lanai gen. et sp. nov. and Ceciamaralia nonatoi gen. et sp. nov., which present unique, external and internal (jaw apparatus), morphological characters.

Materials and Methods

Sampled area

The specimens analyzed were collected in two broad oceanographic campaigns carried out in Brazilian waters (Southwest Atlantic Ocean), coordinated by CENPES/PETROBRAS: (AMBES: Environmental Characterization of the Espírito Santo Basin (18°–21°S/37°–40°W) and HABITATS: Assessment of the Environmental Heterogeneity of the Campos Basin (21°–24°S/38°–45°W) (Lavrado & Brasil, 2010). The collections were done between 2008 and 2013 at depths ranging from 12 to 3,301 m; the organisms were fixed in 4% formalin and then preserved in 70% ethanol.

Morphological analysis

The external morphology of the specimens was analyzed using a ZEISS Axioscop 2 Plus compound microscope and drawings were made with a camara lucida attached to the microscope. The images were captured with a ZEISS AxioCam MRc attached to a ZEISS Axio Imager M2 and Axio Zoom V.16. All images and figures were edited using Adobe® Photoshop and Inkscape® (The letter ‘i’ was skipped from all illustrations to avoid confusion with scale bars of the images).

To perform the scanning electron microscopy (SEM), specimens were dehydrated in an ethanol baths series at the following concentrations: 70% ethanol (5 min), 80%, 90%, 95% (15 min each) and in absolute ethanol, in three changes (15, 30 and 60 min). Critical point drying (Balzers CPD-30) was performed at 37 °C and at a 70 BAR of CO2 gas input, followed by gold coating using SPD-050 sputter coater (Steiner & Santos, 2004). Specimens on stubs were observed in a JEOL JSM-5800 LV scanning electron microscope and images were taken with the software Semafore (v5.2). Critical point drying, gold-coating and SEM analysis were all performed at the Laboratório de Microscopia Eletrônica, Instituto de Biologia, Universidade Estadual de Campinas (LME-IB/UNICAMP).

The jaw apparatus was analyzed using two different methods: i) placing the entire specimens on a drop of Hoyer solution (trichloroacetaldehyde) or Aquatex® on a slide and coverslip, or ii) placing the specimens between the slide and coverslip, waiting for it to dry and analyzing the jaws by tissue transparency (without damaging the specimens and recovering their integrity by putting them back in ethanol). All observations were done using the ZEISS Axio Imager M2 and Axioscop 2 Plus microscopes.

Cladistic analysis

To analyze the relationship of Ceciamaralia gen. nov. with morphologically similar genera of Dorvilleidae, we performed a cladistic analysis utilizing the character matrix and data developed in the study of de Oliveira Bonaldo et al. (2022), which analyzed the cladistic relationships of the following genera: Dorvillea Parfitt, 1866, Eliberidens Wolf, 1986a, Gymnodorvillea Wainwright & Perkins, 1982 Marycarmenia Núñez, 1998, Meiodorvillea Jumars, 1974, Pettiboneia Orensanz, 1973, Protodorvillea Pettibone, 1961 and Schistomeringos Jumars, 1974. We added four new characters to the matrix (characters 43 to 46) and included a new character state for character 40, to accommodate Ceciamaralia gen. nov. (Table 1). We also followed the methodologies of de Oliveira Bonaldo et al. (2022), keeping the characters coded as binary or multistate, coded as ‘−’ when the character is non-applicable and ‘?’ when the state of the character is unknown. All characters are unweighted. The final matrix comprised 21 species (Table 2) including the outgroup composed of Pettiboneia urciensis Campoy & San Martin, 1980, Pettiboneia wui Carrasco & Palma, 2000 (Dorvilleidae) and Ninoe jessicae Hernández-Alcántara, Pérez-Mendoza & Solís-Weiss, 2006 (Lumbrineridae). We incorporate Pettiboneia in the outgroup because in the phylogenetic study by Struck, Purschke & Halanych (2006), Pettiboneia urciensis resulted out of the Dorvilleidae clade and was closely related to Lumbrineridae. The matrix has 46 morphological characters.

Table 1 List of characters and characters states after de Oliveira Bonaldo et al. (2022), on which the present analysis was based.

Character 40: states modified; characters added: 43–46.

Characters	States	
1. Pair of ocelli/eyespot in posterior region of prostomium	0: absent, 1: present	
2. Pair of ocelli/eyespot in anterior region of prostomium	0: absent, 1: present	
3. Antennae	0: absent, 1: present	
4. Shape of antennae	0: simple, 1: biarticulated, 2: more than two articles	
5. Palps	0: absent, 1: present	
6. Shape of palps	0: simple, 1: biarticulated	
7. Length of palps	0: shorter than the first peristomial ring, 1: longer than the first peristomial ring, 2: longer than the second chaetiger	
8. Dorsal cirri in the first parapodium	0: absent, 1: present	
9. Dorsal cirri on anterior region (first parapodia)	0: absent, 1: present	
10. Dorsal cirri on following parapodia (medium and posterior region)	0: absent, 1: present	
11. Shape of dorsal cirri	0: simple, 1: biarticulated, 2: long digitiform	
12. Acicula in the dorsal cirri	0: absent, 1: present	
13. Position of dorsal cirri on the parapodium	0: proximal, 1: middle, 2: distal	
14. Ventral cirri on the first parapodium	0: absent, 1: present	
15. Ventral cirri on following parapodia	0: absent, 1: present	
16. Position of ventral cirri on the parapodium	0: proximal, 1: middle, 2: distal	
17. Pygidial (anal) cirri	0: absent, 1: one pair, 2: two pairs	
18. Shape of the longer pygidial (anal) cirri	0: simple clavate, 1: annulated, 2: filiform, 3: pseudoannulated	
19. Longer pair of pygidial (anal) cirrus	0: dorsal, 1: ventral	
20. Number of supra-acicular chaetae	0: one to three. 1: four or more	
21. Chaeta which accompanies the capillary in supra-acicular fascicle from anterior region	0: capillary, 1: furcate, 2: geniculate, 3: geniculate with short distal-end	
22.Changing in the type of the chaeta which accompanies the capillary in supra-acicular fascicle	0: no, 1: yes	
23. Chaeta which accompanies the capillary in supra-acicular fascicle of medium and posterior region	0: capillary, 1: furcate, 2: geniculate	
24. Cultriform chaeta replacing the furcate in some chaetigers	0: no, 1: yes	
25. Shape of the prong of furcate chaetae	0: blunt, 1: bifid	
26. Number of chaetae in the sub-acicular fascicle	0: three or four, 1: more than four	
27. Tip of the blade of compound chaetae	0: unidentate, 1: bidentate	
28. Thin guard in the tip of compound chaetae	0: absent, 1: present	
29. Distal end of shafts of compound chaetae	0: smooth, 1: serrated	
30. Cultriform or simple chaeta replacing the ventralmost compound in last chaetigers	0: no, 1: yes	
31. Fused teeth in the anterior margin of mandibles	0: absent, 1: present	
32. Free teeth in the anterior margin of mandibles	0: absent, 1: present	
33. Carriers (or carrier-like structures)	0: absent, 1: present	
34. Margin of carriers (or carrier-like structures)	0: smooth, 1: serrated	
35. Shape of superior basal plate	0: straight, 1: pincer-like shape, 2: L shape	
36. Dentition on the margin of superior plate	0: absent, 1: present	
37. Inferior basal plate	0: absent, 1: present	
38. Shape of inferior basal plate	0: straight, 1: pincer-like shape, 2: L shape	
39. Superior row of maxillary plates	0: absent, 1: present	
40. Number of maxillary plates in superior row	0: thirteen or less, 1: more than thirteen, 2: one or two	
41. Inferior row of maxillary plates	0: absent, 1: present	
42. Number of maxillary plates in inferior row	0: thirteen or less, 1: more than thirteen	
43. Extra row(s) of replacement maxillary plates	0: absent, 1: present	
44. Enlarged pharynx/enlarged anterior region	0: no, 1: yes	
45. Ventral cirri present only in few anterior parapodia	0: no, 1: yes	
46. Long spine in the anteriormost free maxillary plate	0: absent, 1: present.	

Table 2 Matrix of taxa and characters based on de Oliveira Bonaldo et al. (2022) with the addition of the two new species described in this study and four new characters (43–46).

‘−’ indicates that the character is non-applicable and ‘?’ that the character is unknown.

	1	2	3	4	5	6	7	8	9	10	11	12	13	14	15	16	17	
Ninoe jessicae	0	0	0	–	0	–	–	0	0	0	–	–	–	0	0	–	2	
Pettiboneia wui	0	0	1	0	1	1	1	0	1	0	2	1	0	?	1	2	2	
Pettiboneia urciensis	1	0	1	0	1	1	1	0	1	0	2	1	0	1	1	2	2	
Gymnodorvillea floridana	0	0	0	–	0	–	–	0	0	0	–	–	–	?	1	1	1	
Marycarmenia lysandrae	0	0	1	0	1	1	0	0	0	0	–	–	–	0	1	1	2	
Dorvillea largidentis	1	1	1	2	1	1	1	0	1	1	1	1	0	?	1	1	2	
Dorvillea clavata	1	1	1	2	1	1	1	0	1	1	1	1	0	1	1	1	2	
Dorvillea cerasina	1	1	1	2	1	1	1	0	1	1	1	?	0	1	1	0	2	
Schistomeringos rogeri	1	0	1	2	1	1	1	0	1	1	1	?	0	1	1	1	2	
Schistomeringos perkinsi	1	1	1	2	1	1	1	0	1	1	1	1	0	1	1	1	?	
Schistomeringos pectinata	1	0	1	2	1	1	1	0	1	1	1	1	0	1	1	1	2	
Protodorvillea orensanzi	1	1	1	0	1	1	2	1	1	1	0	0	2	1	1	2	2	
Protodorvillea bifida	1	1	1	1	1	1	2	1	1	1	0	0	2	1	1	2	2	
Protodorvillea biarticulata	1	0	1	1	1	1	2	1	1	1	0	0	2	1	1	2	2	
Meiodorvillea minuta	0	0	1	0	1	0	0	0	1	0	0	0	1	0	1	1	2	
Meiodorvillea hartmanae	0	0	1	0	1	0	0	0	0	0	–	–	–	0	1	1	2	
Meiodorvillea jumarsi	0	0	1	0	1	0	0	0	1	0	0	0	1	0	1	1	2	
Eliberidens forceps	0	0	1	0	1	0	0	0	0	0	–	–	–	0	1	1	2	
Eliberidens hartmannschroederae	0	0	1	0	1	0	0	0	0	0	–	–	–	0	0	–	2	
Ceciamaralia lanai gen. et. sp. nov.	0	0	1	0	1	0	0	0	1	0	0	1	0	0	0	1	2	
Ceciamaralia nonatoi gen. et sp. nov.	0	0	1	0	?	?	/	0	0	0	–	–	–	0	0	1	2	
	18	19	20	21	22	23	24	25	26	27	28	29	30	31	32	33	
Ninoe jessicae	?	?	?	–	–	–	–	–	–	–	–	–	–	0	0	?	
Pettiboneia wui	0	?	0	1	0	1	0	0	0	0	0	1	0	0	0	0	
Pettiboneia urciensis	?	0	0	1	0	1	0	0	0	0	0	1	0	1	0	0	
Gymnodorvillea floridana	0	–	0	1	1	2	0	0	0	0	0	1	1	0	0	?	
Marycarmenia lysandrae	0	1	0	1	0	1	0	0	0	0	0	1	1	1	0	?	
Dorvillea largidentis	3	0	1	0	0	0	0	–	?	1	1	1	0	1	1	1	
Dorvillea clavata	3	0	1	0	0	0	0	–	?	1	1	1	0	1	1	1	
Dorvillea cerasina	1	0	1	0	0	0	0	–	?	1	1	0	0	1	1	1	
Schistomeringos rogeri	?	0	0	1	0	1	0	0	1	1	0	?	0	1	?	?	
Schistomeringos perkinsi	?	?	0	1	0	1	0	0	?	1	1	1	0	0	0	1	
Schistomeringos pectinata	2	0	?	1	0	1	0	0	1	1	1	1	0	1	1	1	
Protodorvillea orensanzi	2	0	0	1	0	1	1	0	?	1	0	1	1	1	1	1	
Protodorvillea bifida	2	0	0	1	0	1	1	1	?	1	0	1	1	1	1	1	
Protodorvillea biarticulata	?	?	0	1	0	1	0	0		1	0	1	0	?	?	?	
Meiodorvillea minuta	0	0	0	1	0	1	0	0	0	0	0	1	1	0	0	1	
Meiodorvillea hartmanae	0	0	0	1	0	1	0	0	0	0	0	1	1	0	0	1	
Meiodorvillea jumarsi	0	0	0	2	1	1	0	0	0	0	0	1	1	0	0	1	
Eliberidens forceps	0	0	0	3	1	1	0	0	0	0	0	1	1	0	0	1	
Eliberidens hartmannschroederae	0	0	0	3	1	1	0	0	0	0	0	1	1	0	0	?	
Ceciamaralia lanai gen. et. sp. nov.	0	0	0	1	0	1	0	0	0	0	0	1	1	0	0	0	
Ceciamaralia nonatoi gen. et sp. nov.	0	0	0	2	0	2	0	0	0	0	0	1	1	0	0	0	
	34	35	36	37	38	39	40	41	42	43	44	45	46	
Ninoe jessicae	?	–	–	–	–	–	–	–	–	0	0	0	0	
Pettiboneia wui	–	–	–	0	–	1	?	0	–	1	0	0	0	
Pettiboneia urciensis	–	–	–	0	–	1	1	0	–	1	0	0	0	
Gymnodorvillea floridana	?	0	?	0	–	1	0	0	–	0	0	0	0	
Marycarmenia lysandrae	?	0	?	0	–	1	0	0	–	0	0	0	0	
Dorvillea largidentis	1	0	1	1	0	1	1	1	1	0	0	0	0	
Dorvillea clavata	1	0	1	1	0	1	1	1	1	0	0	0	0	
Dorvillea cerasina	?	0	1	1	0	1	1	1	?	0	0	0	0	
Schistomeringos rogeri	?	0	1	1	0	1	1	1	0	0	0	0	0	
Schistomeringos perkinsi	1	0	1	1	0	1	1	1	0	0	0	0	0	
Schistomeringos pectinata	1	0	1	1	0	1	1	1	1	0	0	0	0	
Protodorvillea orensanzi	1	0	1	1	0	1	?	1	?	0	0	0	0	
Protodorvillea bifida	1	0	1	1	0	1	0	1	1	0	0	0	0	
Protodorvillea biarticulata	?	0	?	1	0	1	?	1	?	0	0	0	0	
Meiodorvillea minuta	0	0	0	0	–	1	0	0	–	0	0	0	0	
Meiodorvillea hartmanae	0	0	0	0	–	1	0	0	–	0	0	0	0	
Meiodorvillea jumarsi	0	0	0	0	–	1	0	0	–	0	0	0	0	
Eliberidens forceps	0	1	1	1	1	0	–	0	–	0	0	0	0	
Eliberidens hartmannschroederae	?	2	1	1	2	0	–	0	–	0	0	0	0	
Ceciamaralia lanai gen. et. sp. nov.	–	0	1	0	–	1	2	0	–	0	1	1	1	
Ceciamaralia nonatoi gen. et sp. nov.	–	0	1	0	–	1	2	0	–	0	1	1	1	

The character matrix was assembled using the Mesquite® software (Maddison & Maddison, 2019) and the parsimony analysis was performed using the software TNT® (Goloboff & Morales, 2023), with the heuristic search by the traditional search function starting with 10,000 Wagner trees, utilizing the TBR (tree bisection reconnection) algorithm with 100 replicates and 30 saved trees per replication. We also used TNT® to analyze branch support by standard bootstrap with 1,000 replicates and Bremer absolute support with 40 steps retaining suboptimal trees. Finally, to view and edit the resulting tree we used Winclada® software (Nixon, 2002).

Deposition of specimens

The specimens, SEM stubs and slides, including the type series, were deposited in the Polychaeta Collection (ZUEC-POL) of the Museu de Diversidade Biológica of the Institute of Biology of the Universidade Estadual de Campinas (MDBio - IB/UNICAMP), Campinas, Brazil. Some paratypes were deposited elsewhere in Brazil: Museu de Zoologia of the Universidade de São Paulo, São Paulo (MZUSP) and Museu Nacional do Rio de Janeiro, Rio de Janeiro, Brazil (MNRJ).

The electronic version of this article in Portable Document Format (PDF) will represent a published work according to the International Commission on Zoological Nomenclature (ICZN), and hence the new names contained in the electronic version are effectively published under that Code from the electronic edition alone. This published work and the nomenclatural acts it contains have been registered in ZooBank, the online registration system for the ICZN. The ZooBank LSIDs (Life Science Identifiers) can be resolved and the associated information viewed through any standard web browser by appending the LSID to the prefix http://zoobank.org/. The LSID for this publication is: urn:lsid:zoobank.org:pub:A1EF2E10-4863-49C1-A2E7-CF80BDFE6249. The online version of this work is archived and available from the following digital repositories: PeerJ, PubMed Central SCIE and CLOCKSS.

Results

Taxonomy

Phylum Annelida Lamarck, 1802

Order Eunicida Fauchald, 1977

Family Dorvilleidae Chamberlin, 1919

Genus Ceciamaralia gen. nov.

urn:lsid:zoobank.org:act:22B5ED41-CF25-4A97-8B75-DF336BE1CBE7.

Type species: Ceciamaralia lanai gen. et sp. nov. described herein.

Etymology: Feminine. The genus name “Ceciamaralia” refers to the name Cecília and the surname Amaral of Dr. Antônia Cecília Zacagnini Amaral, a Brazilian researcher who immensely contributed, and still contributes to the enhancement of Annelida knowledge and to the education of zoologists, taxonomists and ecologists, including the three authors of this article.

Diagnosis: Prostomium triangular-shaped with anterior margin rounded. One pair of simple antennae, distally clavate, with a long and slender basal portion. One pair of simple, short and clavate ventrolateral palps, or absent. Two peristomial rings. First two chaetigers usually enlarged to accommodate the large pharynx. First two parapodia shorter than those following and without appendages. Notopodia represented by a large and long dorsal cirrus (with a thin notoacicula) present in a few anterior parapodia or entirely absent. Ventral cirri short and papilliform, present only in a few anterior parapodia. Supra-acicular chaetae: capillary and furcate or geniculate. Sub-acicular chaetae: compound heterogomph falcigers with serrated unidentate blades. Two pairs of clavate pygidial cirri. Jaw apparatus with paired mandibles, medially connected, without fused or free teeth on the anterior margin. Maxillae composed of a posterior ligament fused to a pair of long and serrated basal plates, followed by one pair of anteriormost free maxillary plates with a long thin spine on the anterior margin. Carrier-like structure absent.

Remarks: Ceciamaralia gen. nov. is well distinguished from all other Dorvilleidae genera by: (i) its maxillae composed of a pair of elongated and serrated basal plates and one pair of free maxillary plates with an anterior long and thin spine, (ii) its enlarged pharynx which makes the anterior region of the specimens also enlarged when it is retracted; preserved specimens are found usually with the pharynx protracted out of the mouth, (iii) antennae with a long and slender basal portion and clavate distal end, (iv) first two parapodia slightly shorter and without appendages, and (v) ventral cirri present only in a few anterior parapodia.

The differences between Ceciamaralia gen. nov. and morphologically similar genera of Dorvilleidae are analyzed in detail in the Discussion section.

Key to species of Ceciamaralia gen. nov.

1a) A long, large dorsal cirri present on parapodia 3 to 7–9; furcate chaeta present in supra-acicular fascicle..……………………….……..…… Ceciamaralia lanai gen. et sp. nov.

1b) Dorsal cirri absent; geniculate chaeta present in supra-acicular fascicle………………………..…….……….….……… Ceciamaralia nonatoi gen. et sp. nov.

Ceciamaralia lanai gen. et sp. nov. (Figs. 1–5)

Figure 1 Ceciamaralia lanai gen. et sp. nov., light microscopy.

(A) Complete specimen. (B) Anterior region, dorsal view. (C) Pygidium, ventral view. (D) Parapodia from anterior region, ventral view. (E) Parapodium from posterior region, frontal view. Abbreviations: an: antenna, pa: palp, pr: peristomial rings, vpc: ventral pygidial cirrus, dpc: dorsal pygidial cirrus, dc: dorsal cirrus, vc: ventral cirrus. Scale bars: (A) 200 μm. (B) 50 μm. (C–E) 20 μm.

Figure 2 Ceciamaralia lanai gen. et sp. nov., chaetae, light microscopy.

(A, B) Chaetae from anterior region. (C) Chaetae from posterior region. (D) Sub-acicular compound chaetae. Abbreviations: fc: furcate chaeta, cp: compound chaeta, cap: capillary chaeta. Scale bars: (A–D) 10 μm.

Figure 3 Ceciamaralia lanai gen. et sp. nov., jaw apparatus, light microscopy.

(A–D) Jaw apparatus. Abbreviations: md: mandible, sp: spine, amp: anteriormost maxillary plate, bp: basal plate, lig: ligament. Scale bars: (A–D) 10 μm.

Figure 4 Ceciamaralia lanai gen. et sp. nov.

(A) Anterior region, dorsal view. (B) Anterior region, ventral view. (C) Pygidium, dorsal view. (D) 7th parapodium, anterior view. (E) 19th parapodium, anterior view. (F) Furcate supra-acicular chaeta from anterior region (5th parapodium). (G) Furcate supra-acicular chaeta from posterior region (45th parapodium). (H) Cultriform sub-acicular chaeta (43th parapodium). (J) Compound sub-acicular chaetae (5th parapodium). (K) Mandible. (L) Maxillae. Abbreviations: an: antenna, pa: palps, ph: pharynx, vpc: ventral pygidial cirrus, dpc: dorsal pygidial cirrus, dc: dorsal cirrus, vc: ventral cirrus, sp: spine, amp: anteriormost maxillary plate, bp: basal plate, lig: ligament. Scale bars: (A–C) 25 μm. (D, E) 15.6 μm. (F–H, J–L) 6.25 μm.

Figure 5 Ceciamaralia lanai gen. et sp. nov., scanning electron microscopy.

(A) Anterior region and prostomium, dorsal view. (B) Anterior region, ventral view. (C) Pygidium ventrolateral view. (D) Parapodium from anterior region, ventral view. (E) Parapodia from median region, lateral view. (F) Sub-acicular chaetae from posterior region. (G) Furcate and compound chaeta from median region. Abbreviations: an: antenna, pa; palp, pr: peristomial rings, dpc: dorsal pygidial cirrus, vpc: ventral pygidial cirrus, dc: dorsal cirrus, vc: ventral cirrus, ph: pharynx, cc: cultriform chaeta, mcp: median compound chaeta, dcp: dorsalmost compound chaeta, cp: compound chaeta, fc: furcate chaeta. Scale bars: (A, C, E) 20 μm. (B) 50 μm. (D, F, G) 10 μm.

urn:lsid:zoobank.org:act:3E16785F-8EDD-47E7-8CF4-34D5BD1F4062.

Diagnosis: One pair of palps. Long and large dorsal dorsal cirri with a thin notoacicula present on parapodia 3 to 6–9. Supra-acicular chaetae: capillary and furcate.

Type locality: Off Espírito Santo State, Brazil, 39°10′17.35″W, 19°36′26.24″S, 392 m, muddy.

Type specimens: Holotype: ZUEC-POL 26900 (39°10′17.35″W, 19°36′26.24″S, 392 m, muddy, 14 Dec 2011); Paratypes: ZUEC-POL 26901 (1 specimen, 39°10′17.35″W, 19°36′26.24″S, 392 m, muddy, 14 Dec 2011); ZUEC-POL 26902 (1 specimen, 38°1′8.43″W, 19°34′20.42″S, 450 m, sandy mud, 9 Dec 2011); ZUEC-POL 26903 (1 specimen 39°36′8.52″W, 19°49′7.27″S, 158 m, sandy muddy, 14 Jan 2012); ZUEC-POL 26904 (3 specimens, 39°30′25.23″W, 19°45′54.56″S, 144 m, muddy, 15 Jan 2012); ZUEC-POL 26905 (2 specimens, 39°53′47.1″W, 20°35′16.23″S, 410 m, muddy, 8 Jan 2012; ZUEC-POL 26906 (1 specimen on slide, 38°41′18.43″W, 19°34′20.42″S, 450 m, sandy mud, 09 Dec 2011); ZUEC-POL 26907 (1 specimen on slide, 39°36′9.34″W, 19°49′6.26″S, 181 m, mud, 29 Jun 2013); ZUEC-POL 26908 (1 specimen on slide, 39°30′25.97″W, 19°45′53.43″S, 143 m, muddy, 27 Jun 2013); MZUSP 6463 (2 specimens, 39°53′47.1″W, 20°35′16.23″S, 410 m, muddy, 8 Jan 2012); MNRJP 008066 (2 specimens, 39°53′47.1″W, 20°35′16.23″S, 410 m, muddy, 08 Jan 2012). SEM Material: ZUEC-POL 26909 (1 stub with 3 specimens - 39°53′47.1″W, 20°35′16.23″S, 410 m, muddy, 8 Jan 2012; 40°14′14.08″W, 21°4′4.56″S, 141 m, sandy, 11 Jul 2013).

Etymology: Masculine. The specific epithet “lanai” refers to the surname of Dr. Paulo da Cunha Lana (in memorian), a Brazilian polychaetologist who immensely contributed to the increase of knowledge of Annelida in Brazil and worldwide, and was the supervisor of the senior author of this article.

Description of holotype: Cylindrical body (Fig. 1A). Complete specimen with 46 chaetigers, 4.18 mm long and maximum width of 0.25 mm in the anterior region (0.16 mm in the posterior region), excluding parapodia. First 3–4 chaetigers larger than the rest of the body to accommodate the enlarged pharynx (Fig. 1A). Prostomium triangular-shaped, anterior margin broadly rounded. Ocelli absent. One pair of simple dorsal antennae in the middle of prostomium, distally clavate, with a long and slender basal portion, almost as long as the prostomium (Figs. 1B, 4A and 5A). One pair of simple, ventrolateral, short, and small clavate palps in the base of prostomium, almost half as long as the prostomium (Figs. 1B, 4A, 4B and 5A). Two peristomial rings without appendages, posterior one longer and wider than anterior one (Figs. 1B, 4A, 4B and 5A, 5B).

Parapodia cylindrical, small, and barrel-shaped. First two parapodia smaller than the following, without appendages (Figs. 1B and 5A, 5B). Large and long dorsal cirrus, with a thin notoacicula, almost 2.5 times the length of parapodium, present from chaetigers 3 to 7 (Figs. 1D, 4B, 4D and 5D). Short and papilliform ventral cirrus in the middle of parapodium, from chaetiger 3 to 7 (Figs. 1D, 4B, 4D and 5B, 5D). Following parapodia slightly larger, longer, and without cirri (Fig 1E, 4E and 5E).

Supra-acicular chaetae: one long, thin and serrated capillary (Figs. 2A, 2C) and one furcate with asymmetrical prongs, one slightly shorter and more robust than the other; tip of both prongs blunt (Figs. 2C, 4G and 5G); furcate of first chaetigers with small prongs and prominent serration below the shorter prong (Figs. 2A, 2B and 4F). Sub-acicular chaetae: three compound heterogomph facilgers, slightly different sizes, ventralmost shortest and dorsalmost longest; bifid shafts with a subtle serration on the distal end; short, robust, serrated, and unidentate blades (Figs. 2A–2D, 4J and 5F, 5G). One serrated cultriform chaeta occasionally replacing the ventralmost compound chaeta on the last posterior chaetigers (Figs. 4H and 5F).

Median and posterior regions moniliform. Pygidium truncate and shorter than the previous chaetigers. Two pairs of clavate pygidial cirri, dorsal pair slightly longer than the length of pygidium and ventral pair half the length of the dorsal pair (Figs. 1A, 1C, 4C and 5C).

Paired mandibles medially connected in a region strongly sclerotized; anterior region slightly broader and less sclerotized than the slender posterior region (Figs. 3A–3D and 4K). Maxillae composed of one pair of elongated and serrated basal plates with small uniform sharp teeth on one margin, posteriorly fused to a weakly sclerotized posterior elongated ligament. Basal plates anteriorly followed by one pair of anteriormost free maxillary plates with a long, thin and prominent spine on its anterior margin (Figs. 3A–3D and 4L).

Variation: Complete specimens ranging from 2.9 to 7.6 mm in length and 33 to 61 chaetigers. All specimens ranging from 0.135 to 0.26 mm in maximum width. Dorsal cirri present from chaetiger 3 to 6–9. The presence of ventral cirri usually follows the parapodia in which the dorsal cirrus is present, but in some specimens the ventral cirri can be present in the following one or two parapodia. Cultriform chaetae are occasionally present in posterior chaetigers, but they are also present in the median region of some specimens, and in others they are absent. The enlarged pharynx is characteristic of the genus and usually protracted out of the mouth in preserved specimens (Figs. 4B and 5B); when it is retracted the specimen presents an enlarged anterior region to accommodate the pharynx (Figs. 1A and 4A).

Location and bathymetrics: Off the states of Espírito Santo and Rio de Janeiro, Brazil, 141–450 m, substrates: mud, sandy mud, muddy or sandy.

Remarks: Ceciamaralia lanai gen. et. sp. nov. differs from C. nonatoi gen. et sp. nov. by the presence of large and long dorsal cirri on a few anterior chaetigers and the presence of furcate chaetae in the supra-acicular fascicle. The median and posterior regions of specimens are usually moniliform.

Ceciamaralia nonatoi gen. et sp. nov. (Figs. 6–9)

Figure 6 Ceciamaralia nonatoi gen. et sp. nov., light microscopy.

(A) Complete specimen. (B) Anterior region (specimen with palps), dorsal view. (C) Anterior region, dorsal view (specimen without palps). (D) Posterior end, pygidium ventrolateral view. (E) Parapodium from anterior region, ventral view. Abbreviations: an: antennae, pa: palp, pr: peristomial rings, ph: pharynx, vpc: ventral pygidial cirrus, dpc: dorsal pygidial cirrus, ppr: parapodium from posterior region, vc: ventral cirrus. Scale bars: (A) 200 μm. (B–D) 50 μm. (E) 20 μm.

Figure 7 Ceciamaralia nonatoi gen. et sp. nov., chaetae and jaw apparatus, light microscopy.

(A, B) Chaetae. (C–E) Jaw apparatus. Abbreviations: cap: capillary chaeta, cp: compound chaeta, gc: geniculate chaeta, cc: cultriform chaeta, lig: ligament, sp: spine, bp: basal plate, amp: anteriormost maxillary plate, md: mandible. Scale bars: (F, G) 10 μm.

Figure 8 Ceciamaralia nonatoi gen. et sp. nov.

(A) Anterior region and prostomium, dorsal view. (B) Anterior region, ventral view. (C) Pygidium, ventral view. (D) 4th parapodium, anterior region. (E) 10th parapodium, median region. (F) Geniculate supra-acicular chaeta. (G) Dorsalmost compound sub-acicular chaeta. (H) Median compound sub-acicular chaeta. (J) Cultriform sub-acicular chaeta. (K) Mandible. (L) Maxillae. Abbreviations: an: antennae, ph: pharynx, dpc: dorsal pygidial cirri, vpc: ventral pygidial cirri, vc: ventral cirri, lig: ligament, bp: basal plate, amp: anteriormost maxillary plate, sp: spine. Scale bars: (A, C) 25 μm. (B) 31 μm. (D) 15.6 μm. (E) 10 μm. (F–H, J) 10 μm. (K, L) 6.25 μm.

Figure 9 Ceciamaralia nonatoi gen. et sp. nov., scanning electron microscopy.

(A) Anterior end and prostomium, dorsal view. (B) Anterior end and prostomium, ventral view. (C) Parapodia from anterior region, ventral view. (D) Parapodium from posterior region. (E) Chaetae from posterior region. (F) Supra-acicular chaetae. (G) Chaetae. Abbreviations: an: antennae, pr: peristomial rings, ph: pharynx, vc: ventral cirrus, cc: cultriform chaeta, cp: compound chaeta, gc: geniculate chaeta, cap: capillary chaetae. Scale bars: (A, B) 50 μm. (C–E, G) 10 μm. (F) 5 μm.

urn:lsid:zoobank.org:act:EFF6CD0C-2071-48A2-915D-6F2F8530A343.

Diagnosis: One pair of palps present or absent. Dorsal cirri absent. Supra-acicular chaetae: capillary and geniculate.

Type locality: Off Espírito Santo State, Brazil, 40°12′52.126″W, 21°11′12.073″S, 680 m.

Type specimens: Holotype: ZUEC-POL 26910 (40°12′52.126″W, 21°11′12.073″S, 680 m, 04 Feb 2009). Paratypes: ZUEC-POL 26911 (1 specimen, 40°12′52.126″W, 21°11′12.073″S, 680 m, 04 Feb 2009); ZUEC-POL 26912 (2 specimens, 40°1′55.373″W, 21°47′26.771″S, 780 m, 06 Feb 2009), ZUEC-POL 26913 (3 specimens, 41°18′33.045″W, 23°39′21.880″S, 692.7 m, 28 Jan 2009); ZUEC-POL 26914 (2 specimens 40°26′37.449″W, 22°33′35.143″S, 401 m, 31 Jan 2009); ZUEC-POL 26915 (1 specimen, 40°26′40.289″W, 22°33′33.805″S, 393.4 m, 11 Jul 2008); ZUEC-POL 26916 (1 specimen, 40°17′33.343″W, 22°25′59.389″S, 387.1 m, 31 Jan 2009); ZUEC-POL 26917 (1 specimen, 40°5′18.066″W, 21°44′21.493″S, 401.6 m, 07 Jul 2008); ZUEC-POL 26918 (3 specimens - 39°30′4.65″W, 19°46′34.99″S, 428 m, muddy, 14 Jan 2012); ZUEC-POL 26919 (1 specimen on slide, 40°2′13.825″W, 21°47′26.324″S, 730.5 m, 28 Jun 2008); MZUSP 6464 (1 specimen, 41°18′33.045″W, 23°39′21.880″S, 692.7 m, 28 Jan 2009); MZUSP 6465 (1 specimen, 40°12′52.126″W, 21°11′12.073″S 680 m, 04 Feb 2009); MNRJP 008065 (38°41′19.8″W, 19°34′20.47″S, 449 m, mud, 30 Jun 2013); MNRJP 008064 (1 specimen, 40°1′45.543″W, 22°19′45.730″S, 701.7 m, 30 Jan 2009); MNRJP 008063 (1 specimen, 40°26′37.585″W, 22°33′35.276″S, 400 m, 31 Jan 2009); ZUEC-POL 26920 (3 specimens, 40°2′13,825″W, 21°47′26,324″S, 730.5 m, 28 Jun 2008). SEM Material: ZUEC-POL 26921 (1 stub with 3 specimens, 40°2′13.825″W, 21°47′26.324″S, 730.5 m, 28 Jun 2008 / 40°12′52.126″W, 21°11′12.073″S 680 m, 04 Feb 2009 / 39°30′4.65″W, 19°46′34.99″S, 428 m, muddy, 14 Jan 2012).

Etymology: Masculine. The specific epithet “nonatoi” refers to the surname of Dr. Edmundo Ferraz Nonato (in memorian), one of the greatest Brazilian naturalists and oceanographers who was the pioneer of Brazilian polychaetology, responsible for the education and inspiration of generations of zoologists.

Description of holotype: Cylindrical body (Fig. 6A). Complete specimen with 55 chaetigers, 6.27 mm long and maximum width of 0.41 mm in the anterior region (0.25 mm in the posterior region), excluding parapodia. First 3–4 chaetigers larger than the rest of the body to accommodate the enlarged pharynx (Fig. 6A). Prostomium triangular-shaped, anterior margin broadly rounded. Ocelli absent. One pair of simple dorsal antennae in the middle of prostomium, distally clavate, with a long and slender basal portion, almost as long as the prostomium (Figs. 6B, 6C, 8A, 8B, 9A, 9B). One pair of simple, ventrolateral, short, and small clavate palps on the base of prostomium, almost half as long as the prostomium (Fig. 6B). Two peristomial rings without appendages, posterior wider and longer than anterior (Figs. 6B, 6C, 8A, 8B and 9A, 9B).

Parapodia cylindrical, small and barrel-shaped. First two parapodia smaller than those following, without appendages (Figs. 6B, 8A and 9B). Dorsal cirri absent on all parapodia. Short and papilliform ventral cirri in the middle of the parapodium, from chaetiger 3 to 7 (Figs. 6E, 8D and 9B, 9C). Following parapodia slightly larger, longer and without cirri (Figs. 8E and 9D).

Supra-acicular chaetae: one long, thin and serrated capillary (Figs. 7A and 9F, 9G) and one geniculate with distal region robust and slightly serrated (Figs. 7A, 7B, 8F and 9C, 9F). Sub-acicular chaetae: three compound heterogomph falcigers, almost equal length, ventralmost slightly shortest; bifid shafts with a subtle serration on the distal end; short, robust, serrated and unidentate blades (Figs. 7A, 7B, 8G, 8H and 9E, 9G). One serrated cultriform chaeta occasionally replacing the ventralmost compound chaeta in the last posterior chaetigers (Figs. 7B, 8J and 9C).

Median and posterior regions moniliform. Pygidium truncate and shorter than the previous chaetigers. Two pairs of clavate pygidial cirri; dorsal pair slightly longer than the length of pygidium and ventral pair half the length of the dorsal pair (Figs. 6D and 8C).

Paired mandibles medially connected in a region strongly sclerotized; anterior region slightly broader and less sclerotized than the slender posterior region (Figs. 7C–7E and 8K). Maxillae composed of one pair of elongated and serrated basal plates with small uniform sharp teeth on one margin, posteriorly fused to a weakly sclerotized posterior elongated ligament. Basal plates anteriorly followed by one pair of anteriormost free maxillary plates with a long, thin and prominent spine on its anterior margin (Figs. 7C–7E and 8L).

Variation: Complete specimens ranging from 3.23 to 6.27 mm in length and 46 to 62 chaetigers. Ceciamaralia nonatoi sp. nov. has small and fragile palps, but many specimens do not present them (Figs. 5C and 7A, 7B). The small size of palps and the enlarged pharynx protracted out of the mouth would obscure the scar of a possible broken palp. Therefore, it is debatable whether this is a variation or a methodological problem, so we have diagnosed the species with presence/absence of palps. The ventral cirri are always present, from parapodia 3 to 5–7. Cultriform chaetae are occasionally present in posterior chaetigers, but they are also present in the median region of some specimens, and in some specimens they are absent. The enlarged pharynx is characteristic of the genus and usually protracted out of the mouth in preserved specimens; when it is retracted the specimen presents an enlarged anterior region to accommodate the pharynx.

Location and bathymetrics: Off the states of Espírito Santo and Rio de Janeiro, Brazil, 387.1–780 m deep, substrates: mud or muddy.

Remarks: Ceciamaralia nonatoi sp. nov. differs from its congener by the absence of dorsal cirri and by the presence of a geniculate chaeta instead of a furcate in the supra-acicular fascicle. The variation of the length of the blades of dorsalmost, median and ventralmost compound chaeta is very subtle, while in Ceciamaralia lanai gen. et sp. nov. it is more distinctive. The bathymetric distribution is also a difference between the two species; Ceciamaralia nonatoi gen. et sp. nov. is recorded in deeper waters (387.1–780 m) than Ceciamaralia lanai gen. et sp. nov. (141–450 m).

Cladistic results

The cladistic analysis resulted in one most parsimonious cladogram from 467,210 rearrangements, with best score (length) of 79 steps, consistency index (ci) of 74, retention index (ri) of 87 (Fig. 10). The cladogram shows the monophyly of Ceciamaralia gen. nov., supported by the following synapomorphies: character 40: only one pair of free maxillary plates; character 44: enlarged pharynx/enlarged anterior region; character 45: ventral cirri present only on a few anterior parapodia and character 46: presence of a long and thin spine on the anteriormost maxillary plate. The genera Ceciamaralia gen. nov., Protodorvillea and Dorvillea were well supported by the Bremer absolute support index (9, 14 and 16 respectively) as well as the bootstraps values (87, 92 and 90 respectively) (Fig. 11).

Figure 10 Resulting cladogram demonstrating the monophyly of Ceciamaralia gen. nov. and its relationship with some Dorvilleidae genera based on the study of de Oliveira Bonaldo et al. (2022).

Length of 79 steps, consistency index (ci) 74 and retention index (ri) 87. Number above the circles represents the character and the number below represents the character state. Open circles represent homoplasies and closed circles synapomorphies.

Figure 11 Branches support of the resulting cladogram.

Number above the branch represents the standard bootstrap value and the number below the branch represents the bremer absolute support value.

The inclusion of Ceciamaralia lanai gen. et sp. nov. and Ceciamaralia nonatoi gen. et sp. nov., as well as the addition of four new characters to the matrix of characters in the study of de Oliveira Bonaldo et al. (2022), did not substantially affect the results obtained in the previous study. Ceciamaralia gen. nov. was placed as a sister group of all other genera analyzed, except Eliberidens and Gymnodorvillea, in presenting the synapomorphy of the character 22: the chaeta which accompanies the capillary in the supra-acicular fascicle does not change along the body.

Discussion

At first glance, Ceciamaralia gen. nov. specimens are hard to differentiate from other small-sized dorvilleids, but a closer look reveals their morphological differences and unique morphology. Below, these differences are discussed with some morphologically similar genera, specifically those present both in the cladistic study of this work and in de Oliveira Bonaldo et al. (2022).

Prostomial appendages

Ceciamaralia gen. nov. presents a cylindrical and small-sized body, with small body appendages and a triangular prostomium, as in Protodorvillea, Meiodorvillea, Eliberidens, and Pettiboneia. Those genera also appear closely related in cladistic studies (Eibye-Jacobsen & Kristensen, 1994; de Oliveira Bonaldo et al., 2022). Protodorvillea has long and biarticulated palps, while Ceciamaralia gen. nov. has simple, small, clavate and papilliform palps, when present. The palps of Pettiboneia are shorter than in Protodorvillea but are still biarticulated and also longer and larger than the palps of Ceciamaralia gen. nov. The small clavate palps in Ceciamaralia gen. nov. are similar to those observed in Meiodorvillea and Eliberidens. The antennae are described here as simple and clavate, as in some Dorvilleidae genera, but, in Ceciamaralia gen. nov. they are unique in having a longer and slender basal portion than the antennae from other genera.

Parapodial appendages

Ceciamaralia gen. nov. presents small papilliform ventral cirri only on a few anterior parapodia, while other genera such as Meiodorvillea, Protodorvillea, Pettiboneia, Dorvillea, Schistomeringos and Eliberidens present it on all parapodia, except the first; on the other hand, Eliberidens hartmannschroederae Hilbig, 1995 does not have ventral cirri.

Pettiboneia and Ceciamaralia gen. nov. also share the presence of dorsal cirri on anterior parapodia inserted at the base of parapodia, but they have two evident differences: i) Ceciamaralia lanai gen. nov. presents the dorsal cirri from parapodium three reaching the 9th, while in Pettiboneia they are present from parapodium two reaching at least the 7th, but in some species they can reach as far as the 25th, as in Pettiboneia sanmartini Aguirrezabalaga & Ceberio, 2003; ii) Ceciamaralia lanai gen. et sp. nov. has very long and large dorsal cirri, reaching more than three times the length of parapodia, while in Pettiboneia they are distinctively slender and shorter. Some species of Meiodorvillea, such as Meiodorvillea minuta (Hartman, 1965), also present dorsal cirri in few anterior parapodia, but they are small, papilliform/globular, from the 2nd parapodium and inserted in the middle of the parapodium. Dorvillea and Schistomeringos also present cylindrical dorsal cirri, but they are slender, biarticulated and absent only on the first parapodium. In contrast, Ceciamaralia nonatoi gen. et sp. nov. does not have dorsal cirri.

Dorsal cirri x notopodium x notopodial lobe x branchiae

The presence of the dorsal cirri in Ceciamaralia lanai gen. et sp. nov. generated a debate regarding the origin of this appendage. It resembles the same structure observed in species of Pettiboneia, Diaphorosoma Wolf, 1986a and Westheideia Wolf, 1986a, but they are named differently. All species of these genera present this cylindrical appendage inserted at the base of the parapodia. In Diaphorosoma magnavena Wolf, 1986a and Westheideia minutimala Wolf, 1986a, it is described as a notopodium bearing an internal acicula, and the former having an internal vascular loop, similar to a branchia. It is important to note that both species also present an appendage described as branchia inserted distally on the neuropodium and it also presents a vascular loop as in D. magnavena. The notopodium in Pettiboneia species is described as a dorsal cirrus, also having internal acicula; some species, like P. dibranchiata (Armstrong & Jumars, 1978), also have a distal appendage in the neuropodium described as branchia, exactly as in D. magnavena and W. minutimala. The notopodium of Ceciamaralia lanai gen. et. sp. nov. shows a vascularized tissue and an acicula barely visible, so we decided to describe it as a dorsal cirrus because of its position and in agreement with how it is described in the literature.

Chaetae

The presence and format of furcate and geniculate chaeta shows a great diversity in Dorvilleidae. Of the two species of Ceciamaralia gen. nov., C. lanai gen. et sp. nov. has furcate chaetae, while C. nonatoi gen. et sp. nov. has geniculate chaetae. This variation can also be observed in species of Meiodorvillea; M. minuta possesses furcate and M. apalpata possesses geniculate chaetae, while M. penhae and M. jumarsi present both types. Dorvillea and Schistomeringos are two similar genera, the former lacking furcate while the latter has them. All species of Protodorvillea and Eliberidens present furcate chaeta.

The blades of the compound chaeta of Ceciamaralia gen. nov. are smaller, straighter and more robust than in species of other genera in which the dorsalmost compound chaeta can be very long and spinigerous.

Jaw apparatus

The jaw apparatus of Ceciamaralia gen. nov. presents a distinct and specific morphology differing from that of all other species of the family. Protodorvillea, Dorvillea and Schistomeringos present a broad and robust jaw apparatus with a maxillae composed of strong basal plates, a carrier-like structure and four rows of many robust maxillary plates. The maxillae of Pettiboneia and Meiodorvillea are smaller, presenting only two rows of similar maxillary plates (species of Pettiboneia have some poorly sclerotized additional plates and they lack basal plates). On the other hand, the maxillae of Eliberidens do not have maxillary plates at all; they are composed only of superior and inferior long basal plates. The jaw apparatus of Ceciamaralia gen. nov. presents the posterior ligament fused to only one pair of long and serrated basal plates followed anteriorly by one pair of free maxillary plates presenting a long and thin distinct spine.

Cladistic analysis

The scarcity of taxonomic knowledge of Dorvilleidae is an obstacle to conducting phylogenetic analysis. But some studies have been performed to elucidate relationships within the family; the most comprehensive was a cladistic one carried out by Eibye-Jacobsen & Kristensen (1994) where they utilized all known genera of Dorvilleidae at that time as terminal taxa. Even with slow progress, molecular data is already aiding in the clarification of the phylogenetic relationships of dorvilleids, mainly Ophryotrocha (Kvalø Heggøy, Schander & Åkesson, 2007), which is the genus with most sequence data. The lack of molecular data for other genera of the family opens space for morphological cladistic studies like Pleijel & Eide (1996), de Oliveira Bonaldo et al. (2022) and this present one. Those studies are important to provide data and results for future studies on the systematics of Dorvilleidae.

Ceciamaralia gen. nov. morphologically resembles other small-sized dorvilleids presented in the cladistics analysis by de Oliveira Bonaldo et al. (2022); hence we included both new species described here in the matrix of that study. The new genus appeared as monophyletic based on the specific synapomorphies discussed here: the unique maxillae with only one pair of free maxillary plates, presenting a specific long and thin spine, the enlarged pharynx making the anterior region enlarged when it is retracted, which is not observed in others genera of the family, and the ventral cirrus present only in few anterior parapodia. The results of de Oliveira Bonaldo et al. (2022) placed Meiodorvillea as a sister group of all other genera presented in the analysis except Eliberidens and Gymnodorvillea. The inclusion of the Ceciamaralia gen. nov. species and the new characters in the analysis did not affect the previous relationship results among the genera or their monophyly.

Present and future

The study of small annelids has some obstacles like the difficulty to collect and identify them. In Brazil only nineteen species of Dorvilleidae were registered before the present study, but this number does not reflect the true diversity of this family on the Brazilian coast. The continuous increase of scientific advancements and the development of new techniques and tools, researchers can perform new and more detailed analyses of unidentified species. These studies increase the systematic knowledge of the species and reveal the biodiversity of the group.

Museum collections play an important role as they are a depository for the types of previously described organisms and also contain unidentified organisms, which can hold much biological and ecological information aiding in several fields of study, mainly taxonomy and ecology. The specific identification of the organisms reveals records and occurrences of them aiding in biogeographical, ecological and distribution studies and ecological patterns subsidizing data of potential distribution (Budaeva et al., 2024). Morphological analysis can reveal new or different characters and structures supporting a refined description, reveal new species and aid the understanding of the phylogenetic relationship of the species of the group, as was demonstrated in the present study.

The incentive towards taxonomic studies and projects resulted in the first description of a new genus of Dorvilleidae in almost 25 years, presented here. Dorvilleids present a great morphological diversity, but our taxonomic knowledge of this group is still limited by the reasons mentioned above and the lack of incentive for taxonomist studies. This incentive is very important to aid researchers to better comprehend and classify those organisms.

Preliminary morphological analysis of museum materials of Dorvilleidae indicates several new records of the family for the Brazilian coast and also potential new species for the family. In addition, we highlight the importance of the effort to collect new and fresh organisms in view of the fact that they can provide current biodiversity data and can also provide more accurate genetic information through molecular studies, particularly because groups like Dorvilleidae present a huge gap in those data.

Supplemental Information

Supplemental Information 1 Cladistic Analysis Files.

We would like to thank all people involved in the collection of the material (projects AMBES and HABITATS) and also the MDBio for providing access to it. We would like to acknowledge the access to equipment and assistance provided by the Electron Microscope Laboratory (LME/UNICAMP). We thank Dr. Yasmina Shah Esmaeili for providing language revision. We also express our gratitude for the three reviewers, Dr. Vinicius Miranda, Dr. Danny Eibye-Jacobsen and one anonymous, for all the corrections and suggestions which immensely contribute to the improvement of this work.

Additional Information and Declarations

Competing Interests

Author Contributions

Data Availability

New Species Registration

The authors declare that they have no competing interests.

Rafael de Oliveira Bonaldo conceived and designed the experiments, performed the experiments, analyzed the data, prepared figures and/or tables, authored or reviewed drafts of the article, and approved the final draft.

Tatiana Menchini Steiner analyzed the data, authored or reviewed drafts of the article, and approved the final draft.

André Rinaldo Senna Garraffoni analyzed the data, authored or reviewed drafts of the article, and approved the final draft.

The following information was supplied regarding data availability:

The metafiles of the cladistic studies like the nexus and TRE files are available in the Supplemental Files.

The following information was supplied regarding the registration of a newly described species:

Publication LSID: urn:lsid:zoobank.org:pub:A1EF2E10-4863-49C1-A2E7-CF80BDFE6249

Ceciamaralia LSID: urn:lsid:zoobank.org:act:22B5ED41-CF25-4A97-8B75-DF336BE1CBE7;

Ceciamaralia lanai LSID: urn:lsid:zoobank.org:act:3E16785F-8EDD-47E7-8CF4-34D5BD1F4062;

Ceciamaralia nonatoi LSID: urn:lsid:zoobank.org:act:EFF6CD0C-2071-48A2-915D-6F2F8530A343.

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
