# Peer review of "Ceciamaralia, a new genus of Dorvilleidae (Annelida) from deep waters of the Southwest Atlantic Ocean and an insight into its relationship within the family"

_PeerJ, doi:10.7717/peerj.18358_

## Round 0.1 · original submission · Major Revisions

Dear Authors,

Please find attached three reviews of your ms describing a new genus of dorvilleid from the SW Atlantic.

The reviewers have each identified several points that need to be addressed. I concur with their findings, in particular the following:

1. More clear/convincing arguments for the unique shape of the maxillae and basal plates and carriers;
2. Provide a table of characters used, rather than refering to previous work - this can be in the main text in my view, but at the very least in a Supplementary file
3. A more uniform treatment of character descriptions, and a discussion of homology as well as any variation observed
4. In regard to the cladogam, one reviewer suggests to account for/discuss all character steps (currently only 61 of 83 are documented); I think this is a good idea, but if you find this unnecessary, please say why;
4. and lastly, please pay attention to the English, especially in the descriptive material (standardize on either telegraphic style OR complete sentences NOT a combination of both);

Of course, you are free to refute any of the reviewers suggestions, but I note that quite a few suggestions are shared by two or more reviewers, which gives their view more credence.

I look forward to your responses and revised ms.
All the best, Chris

Reviewer 1 ·

Basic reporting

MS by Rafael de Oliveir a Bonaldo, Tatiana M Steiner and André R S Garraffoni entitled Ceciamaralia, a new genus of Dorvilleidae (Annelida) from deep waters of the Southwest Atlantic Ocean and an insight into its relationship within the family
is devoted to the description of species and genus new to science of one of the most mysterious and poorly known families of marine Annelids - Dorvilleidae. The article describes new findings of dorvilleids from relatively deep-sea areas of the Brazilian coast. Thus, the topic of the paper is relevant and interesting. Descriptions of new taxa generally correspond to modern requirements for taxonomic descriptions. The MS may be accepted for publication after serious revision.

Experimental design

See below

Validity of the findings

See below

Additional comments

Specific comments on the results

1) Some photographs are of very poor quality, primarily this concerns photographs of the maxillary apparatus (Fig 2, Fig 5). The drawings of the maxillary apparatus (Fig 3L and Fig 6L) do not entirely correspond to what is seen in the photographs.
2) It is not clear from the photographs how correctly the authors interpret the shape of the maxillae armed with spines; in my opinion, this may just be a thickened part of the flat maxillary plate. In this case, the diagnosis of the genus will have to be changed. In my opinion, for a definite judgment it is necessary to take an SEM photo of the maxillary apparatus, or at least obtain photographs of good quality and resolution. This seems is possible given the large number of specimens of each species.
3) In FIG. 6 L basal plates are asymmetrical, is this true, or the figure shows the basal plate bent as an artifact of fixation
4) Maxillae II and III are shown in the figures with the same row of small denticles. Does this correspond to the facts, or do these denticles still alternate in size? The fact is that most Dorvilleidae have an alternation of large and smaller teeth on the maxillae - this is due to their feeding habits.
5) The authors describe a “ligament” that unites the right and left rows of maxillae, but this is a typical carrier-like structure; it is unclear why new terms need to be introduced.
6) The authors write about a muscular pharynx of increased volume (compared to other dorvilleids), but there are no images of such a position of the pharynx and, most importantly, there are no drawings or photographs of the position of the jaw apparatus, especially the most anterior maxilla with a long spine.

Specific notes for discussion.
7) The authors do not make any assumptions about how similar spines on maxilla II can be used for feeding. The fact is that nothing similar is known in other dorvilleids.
8) The authors use cladistic methods to search for the position of the new taxa they described. Perhaps they have no other choice, since the material on which the work is based is fixed with formaldehyde and therefore almost unsuitable for molecular genetic methods. However, since they are using this method, it is necessary to provide the entire table of characters used and not refer to the previous work of the co-authors, which provides the corresponding table of characters, see (Table 1(on next page) List of characters added (44-47) and states of characters modiûed (41) to the list of characters of the study of de Oliveira Bonaldo et al., (2022) on which the present analysis was based). It is impossible to understand the table of characteristics, and it is impossible to understand the “tree” if 80% of the characteristics must be looked for in another publication. By the way, this one does not use the sign of alternating teeth of different sizes in the maxillae. And this can be significant.

8) The authors make no attempt to compare the phylogenetic tree they obtained with existing phylogenetic schemes, for example, with the scheme proposed in Tilic & Rouse, 2024, and based on currently available molecular data.

·

Basic reporting

The article has its merits in describing a new genera and two new species of Dorvilleidae. Specially in a group of small dorvilleids, in which the morphology seems quite uniform. Yet, the authors managed to find good support to their findings and to erect the new genus and species. I also would like to highlight that they are doing a good work, not only describing the diversity find along the Brazilian coast, but also in revising some, long forgotten, taxa within Dorvilleidae - and here I highlight the fact brought in their conclusions, that studies about dorvilleids is too focused in those species belonging to Ophryotrocha.
However I must say that I strongly believe that their description of the species can be reformulated and some characters better described and explained. Certain characters are described too superficially, while others are described in a way that doesn't reflect what is illustrated (I made comments in the .doc file so each case is highlighted there).
I also advocate for a proper language in the species description. Usually the telegraphic form suits better for species description. I call attention to this, as in several occasions the use of adverbs (instead of a clear, direct, and telegraphic language) let the description dubious or confuse.
The literature used is ok to what they proposed. There is an incongruent data regarding the species diversity in Brazil and the reference listed for this subject, but this is easy to correct. And I believe that if they improve their discussion (see comments below), then the number of references might increase as well.
All shared material, and illustration are well prepared and suitable to understand their results.
As I pointed in the .doc file, I believe that the remarks and the discussion could be improved. Regarding the new Genus Remarks, I believe that all the remarks should encompass the variation in some character within the genus, and also to compare the new genus with remaining genus of the family. To achieve this, I suggest the model used by Marian Pettibone (see Pettibone - 1992 - Two New Genera and Four New Combinations of Sigalionidae (Polychaeta)). About the discussion at the end of the document, then I believe the discussion of the characters can be improved in better delineating why they discuss only with 7 of the 31 genera of Dorvilleidae, or by also discussing and arguing having the remaining genera of the family. As the title of the article suggest "insight in its relationship within the family", then I think: how can an insight at a family level exist, if only 1/4 of the family was taken in consideration to construct such insight? (I also made comments about this through the document).

Experimental design

The research agrees with the journal aim and scope. The research was performed following the good practices according to the ethical standard.

About the methodology, I missed some information. It is ok to me that they followed the Character Matrix presented in de Oliveira Bonaldo (2022) however, they included more data in the matrix, in a way they have a modified matrix, so I suggest to include the new matrix as an supplementary file within this work. It can be problematic to a reader having to start observation in one paper, them finish the observation in another. I strongly believe that all information regarding the experiment should follow the paper in which is describe the results - and so is the case of the cladistic analysis of the new genus.

Another issue is the absence of a description of the characters, in a way we can understand their hypothesis of homology of each character and how they vary in the taxa used in the cladistic analysis.

Other comments regarding the methodology can be seen in the comments made trough the document attached.

Validity of the findings

The results are valid and self supporting. All available data are sufficient to make a re-analysis and I found the same results as the authors did - the only exception is that I found a different number of trees than the authors, but this can be a bias of the computer memory.

The conclusions are well stated and in agreement to what was proposed as subject of research.

Additional comments

no comments.

·

Basic reporting

The scientific content of this manuscript is generally of a high quality. I comment on this below.

There are, however, three areas in which this paper is problematic:

1) There are a large number of problems with the quality of English used. I have suggested a couple of hundred corrections in the attached PDF, but a thorough revision of the language is necessary. This is rather surprising, as at least two of the authors are very experienced.
Somewhat related, the species descriptions are a strange mixture of narrative and telegraphic styles. This is inconsistent, and I would assume that the journal would prefer a clear telegraphic style.

2) In many instances the formatting is - and I hesitate to use this expression - sloppy. Several examples are given in the attached PDF, but just one further example of the inconsistencies seen is given here; it pertains to the list of references (which I did not feel the need to correct in detail). In the individual references the name of the journal is followed by a period (dot), a comma, or neither, apparently completely at random. This tells me that the authors have not proofread their own manuscript.

3) This point is more difficult to describe, but I will do my best. I found that the manuscript contains several phrases that "sound good", but really are either exaggerated or uninformative (what the Americans might call "fluff"). Six examples follow:
a) Abstract, line 22: "many gaps regarding its [Dorvilleidae's] diversity"
b) Abstract, lines 24-25: "unveiling the hidden diversity of this family" [note: this is repeated on line 84]
c) Abstract, lines 35-36: "This study is one of several recent..." [probably referring to de Oliveira Bonaldo, Steiner & Amaral 2022 and de Oliveira Bonaldo et al. 2022]
d) Discussion, line 429: "The scarcity of knowledge..."
e) Discussion, lines 437-438: "Those studies aid not only..."
f) Discussion, lines 457-474: Two paragraphs (!) that are practically without tangible content.

Experimental design

The authors have presented the description of a new genus of dorvilleid annelids, with two new species, from Brazil. The descriptions are convincing and are based on a considerable number of specimens.

It is, however, strange that the authors provide a list of characters (lines 178-183) that distinguish Ceciamaralia gen. nov. from other genera of Dorvilleidae, but that this list does not contain the fact that ventral cirri are only present on a few anterior chaetigers, even though the authors later define it as a character (no. 46) in their phylogenetic analysis.

The figure are quite good. A note on the figure legends: for Fig. 3D-E the view of the parapodium is given as "lateral", which can be either "anterior" or "posterior". It would be preferable to use one of the latter terms for precision. Furthermore, for Figs 6D-E and 7D no information is given on the perspective.

Validity of the findings

The discussion provides a full comparison of the new genus to other dorvilleid genera and also contains a phylogenetic analysis that supports the monophyly of Ceciamaralia gen. nov.

However, the cladogram shown on Fig. 8 is incomplete in the sense that not all characters are mapped on it. Thus, for a total length of 83 steps, only 61 steps are documented.

---

## Round 0.2 · Minor Revisions

Dear Author, please find attached further suggestions as part of the Round 2 review process for the improvement of your MS. 'Reviewer 3' has provided some highly relevant suggestions that should be considered. In addition, I have provided a 'technical edit' for your consideration. Please revise your MS using both reports and submit a revised copy with track changes. If there are any points that you disagree with then these can be explained in a covering letter. All the best, Chris

·

Basic reporting

This is a review of the resubmitted version of manuscript 99857 (de Oliveira Bonaldo, Steiner & Garraffoni). I find this revised version greatly improved compared to the original submission and was pleased to see that most of my suggestions for improvement have been followed.
In particular as regards issues of language (grammar, spelling, etc.), however, I must emphasize that the corrections I provided to the original version were (as stated then) far from complete. It was my hope that those corrections would lead the authors to thoroughly review the text and make further necessary changes. This may have taken place, but it must be concluded that much work still remains to bring the language up to a level that is acceptable in a prestigious international journal. The editor has indicated that this will take place, and I have great confidence in him, but the extent of this task should not be underestimated.
Apart from this, the improvements made are considerable. I would especially like to thank the authors for providing further images of the jaw apparatus in Ceciamaralia lanai (Fig. 3) as well as in C. nonatoi (Fig. 7). I have a few further remarks (see below) and assuming that they are dealt with, I believe this manuscript can be accepted for publication.

Experimental design

No comment.

Validity of the findings

More specific remarks:
1) In line 93 the authors write that they have applied "an integrative approach", but this term is typically used for studies that include both morphological and molecular evidence, which is not the case here.
2) In lines 139–141 the authors refer to Pettiboneia urciensis and P. wui as outgroups, along with Ninoe jessicae. In the original submission, only Ninoe jessicae was specified – and the two species of Pettiboneia are certainly not outgroups!
3) In the remarks for Ceciamaralia gen. nov. (lines 193–199) the fact that the ventral cirri only occur on a few anterior parapodia is not mentioned as a distinguishing character for this new genus. It seems strange to overlook such an obvious character. A similar statement is provided regarding the dorsal cirri, but it should be remembered that these cirri are completely absent in C. nonatoi sp. nov.
4) There is a risk that the remarks for C. nonatoi (lines 344–346) will cause confusion. The term "variation of the length of the blades of compound chaetae" refers to the length variation within a given parapodium, which is greater in C. lanai than in C. nonatoi, but it could easily be misinterpreted to refer to the length variation of these chaetae throughout the entire animal.
5) On line 446, I suggest that the word "subside" should probably be "provide".
6) In the legend for Fig. 5 the abbreviation pr (peristomial ring(s)) is listed twice.

Additional comments

Finally, I must add that the list of references is still quite chaotic. It doesn't follow any perceptible formatting principles. Just as an example, a comma is used to separate journal name + volume number from the page numbers in 23 references, whereas a colon is used in 14 references. In other cases the volume number is missing entirely, abbreviations are used for many journal names (which most journals no longer accept), italization is not used consistently, etc., etc. Please choose a format, preferably the one used by the journal, and apply it consistently to all references.
Once all these issues are solved, I look forward to seeing this paper in print.
Best regards,
Danny

---

## Round 0.3 · Minor Revisions

Dear Author, Thank you for dealing with the comments and suggestions of the Reviewer in the 2nd round of review. I agree with all of your responses, except for the one concerning the use of the Pettiboneia species as outgroups. This needs justification in the MS, as it is somewhat counter-intuitive to include members of the study family in the outgroup. Also, please find attached my further suggestions for the improvement of your MS (some of these are repeated from in my earlier review, as you were not able to read the track changes comments - this time the comments are placed in square brackets inside the text). Please revise your MS and submit a revised copy with track changes. If there are any points that you disagree with then these can be explained in a covering letter. All the best, Chris

---

## Round 0.4 · accepted · Accept

Dear Author, thanks for your patience during the review process and for incorporating the suggestions in the review. I am happy to accept the MS now. Looking forward to seeing it published, best, Chris